# Immune parameters of HIV susceptibility in the female genital tract before and after penile-vaginal sex

Avid Mohammadi [1✉], Sareh Bagherichimeh[1,5], Yoojin Choi[2], Azadeh Fazel[1], Elizabeth Tevlin[3,6], Sanja Huibner[1], Sara V. Good[4], Wangari Tharao[3] & Rupert Kaul [1✉]

## Abstract

**Background** In women, most HIV infections are acquired through penile-vaginal sex. Inflammation in the female genital tract (FGT) increases the risk of HIV acquisition and transmission, likely through recruitment of HIV target cells and disruption of epithelial barrier integrity. Although sex may have important immune and epithelial effects, the impact of receptive penile-vaginal sex on the immune correlates of HIV susceptibility in the female genital tract is not well described.

**Methods** STI-free heterosexual couples were recruited to the Sex, Couples and Science (SECS) Study, with the serial collection of cervical secretions (CVS), endocervical cytobrushes, blood and semen before and up to 72 h after either condomless ($n = 29$) or condom-protected ($n = 8$) penile-vaginal sex. Immune cells were characterized by flow cytometry, and immune factors including cytokines and soluble E-cadherin (sE-cad; a marker of epithelial disruption) were quantified by multiplex immunoassay. Co-primary endpoints were defined as levels of IP-10 and IL-1α, cytokines previously associated with increased HIV susceptibility.

**Results** Here we show that cervicovaginal levels of vaginal IP-10, sE-cad and several other cytokines increase rapidly after sex, regardless of condom use. The proportion of endocervical HIV target cells, including Th17 cells, activated T cells, and activated or mature dendritic cells (DCs) also increase, particularly after condomless sex. Although most of these immune changes resolve within 72 h, increases in activated cervical CD4 + T cells and Tcm persist beyond this time.

**Conclusions** Penile-vaginal sex induces multiple genital immune changes that may enhance HIV susceptibility during the 72 h post-sex window that is critical for virus acquisition. This has important implications for the mucosal immunopathogenesis of HIV transmission.

## Lay Summary

Women who acquire HIV most commonly do so during penile-vaginal sex. Although the risk of HIV acquisition is higher when there is pre-existing inflammation in the female genital tract, the impact of receptive penile-vaginal sex itself on immune markers of HIV susceptibility in the genital tract has not been widely studied. We recruited heterosexual couples, without HIV or sexually-transmitted infections, and studied the impact of a single episode of penile-vaginal sex on immune cells and proteins in the female genital tract. We found that some markers within the cervix and vagina increased immediately after sex, then returned to normal. We noticed differences in these changes depending on whether the sex was condom-protected and whether the male partner was circumcised. Our findings might help us to understand how sex impacts the immune system and how this might contribute to HIV acquisition.

[1] Department of Medicine, University of Toronto, Toronto, ON, Canada. [2] Department of Immunology, University of Toronto, Toronto, ON, Canada. [3] Women's Health in Women's Hands Community Health Center, Toronto, ON, Canada. [4] Department of Biology, University of Winnipeg, Winnipeg, MB, Canada. [5]Present address: Department of Pathology and Laboratory Medicine at Schulich Medicine and Dentistry, University of Western, London, ON, Canada. [6]Present address: Street Health Community Nursing, Toronto, ON, Canada. ✉email: Avid.mohammadi@mail.utoronto.ca; Rupert.kaul@utoronto.ca

Sexual transmission accounts for the majority of incident global HIV infections, and women are more susceptible to heterosexual HIV acquisition than men[1]. This increased risk in women relates in part to socio-economic factors and potential exposure of the more susceptible anorectal mucosa during receptive anal sex[1], but vaginal acquisition risk is higher than penile and can be enhanced by factors that induce inflammation in the female genital tract such as sexually transmitted infections (STIs), bacterial vaginosis (BV) and vaginal washing[2–5].

A mucosal inflammatory response provides an important host immune defense against many infectious pathogens. However, inflammation in the genital tract increases the risk of HIV acquisition, both by impairing the integrity of the epithelial barrier and by recruiting HIV-susceptible target cells[6–9]. Preferential cell targets for HIV during mucosal virus acquisition include activated CD4 + T cells that express the HIV co-receptor CCR5, particularly Th17 (CCR6 + ) cells, as well as dendritic cells (DCs) and Langerhans cells that play a critical role in virus dissemination in the genital tissue as well as to secondary lymphoid organs[10–21]. Inflammation attracts HIV target cells by inducing chemokines that guide immune cells to the site of inflammation, such as the recruitment of neutrophils by CXCL8 (IL-8), of activated CD4 + T cells by CCL3, CCL4 and CXCL10 (MIP-1α, MIP-1β and IP-10), and of Th17 cells by CCL-20 (MIP-3α)[22–28]. In addition, the inflammatory cytokines TNF and IL-1α can directly disrupt cell-cell junctions and reduce epithelial barrier integrity[29,30]. In keeping with this, a higher genital level of these chemokines/cytokines has been associated with increased HIV acquisition risk in both women and men[6,31], and proteome analysis of vaginal secretions has demonstrated associations between inflammatory cytokines that reduce barrier integrity, neutrophil proteases such as MMP9, and the mucosal density of HIV target cells[8,9].

Although these mucosal immune factors correlate with HIV susceptibility in human cohorts, participant sampling in such studies is generally performed after a period of sexual abstinence, while virus penetration of the genital mucosa occurs within hours of sexual contact[32–34]. Therefore, it is critical to understand how penile-vaginal sex affects the mucosal immunology and epithelial barrier integrity of the female genital tract (FGT). Mechanical aspects of penile-vaginal sex may impose physical stress on the epithelium, and condomless sex results in vaginal exposure to semen, which is rich in both pro-inflammatory and regulatory cytokines[35,36]. In vitro and small animal studies demonstrate that vaginal epithelial cells respond to seminal plasma by up-regulating pro-inflammatory cytokines/chemokines and recruiting leukocytes to the FGT within a few hours[37–39]. However, there are fewer data from human studies regarding the mucosal impact of penile-vaginal sex, and published studies provide conflicting results. While one study reported recruitment of immune cells (APCs, T cells) and elevated expression of pro-inflammatory cytokines/chemokines (IL-1α, IL-6, IL-8 and GM, CSF) within 12 h of condomless penile-vaginal sex[40], other work demonstrated a decrease in mucosal innate immune responses 2–6 h after sex, with reduction in IL-8, antimicrobial peptides, and anti E. coli activity of cervical secretions[41].

The goal of the Sex, Couples and Science (SECS) study was to define the immediate and short-term impacts of condomless and condom-protected penile-vaginal sex on the genital immune correlates of HIV susceptibility. Specifically, we hypothesised that condomless penile-vaginal sex would induce transient inflammation in the FGT that would resolve within 48–72 h after sex, the typical period of abstinence that is requested in many clinical studies of genital immunology and HIV susceptibility. Our results demonstrate that receptive penile-vaginal sex induces rapid alterations in cervico-vaginal proinflammatory cytokines, endocervical cell populations and epithelial integrity as early as 1 h after sex. Although most changes resolve within 72 h, they may have important effects on host HIV susceptibility within this critical window of virus acquisition.

## Methods

**Study design.** Couples were recruited into this prospective observational cohort study through the Women's Health in Women's Hands Community Health Center (WHIWHs) in Toronto as described previously[42]. The study protocol was approved by the HIV Research Ethics Board at the University of Toronto. Flyers were posted within the WHIWHs centre and across the University of Toronto St. George campus to recruit participants. Prior to recruitment, the research nurses at WHIWH provided a detailed overview of the study details and requirements to potential participants. At the pre-screening visit, written informed consent was obtained from all participants, and they were tested for sexually transmitted infections and pregnancy. Exclusion criteria were infection with HIV1/2, syphilis, *Neisseria gonorrhoeae* (GC) and/or *Chlamydia trachomatis* (CT); Ag <16 years; pregnancy; any genital ulcers or discharge; irregular bleeding; taking immunosuppressive medications and having taken antibiotics within one month prior to study enrollment.

**Study approval.** The protocol was approved by the HIV Research Ethics Board at the University of Toronto (ethical approval number: 33381). At the screening visit, the research nurses at WHIWH provided detailed information about the study to potential participants and written informed consent was taken from all interested participants.

**Sampling protocol.** The study protocol consisted of four visits: screening, baseline (48 h before sex) and follow up visits, i.e., 1–2 h, 72 h post-sex (Fig. 1). At the screening visit, blood and urine were collected for STI diagnostics. Eligible participants were

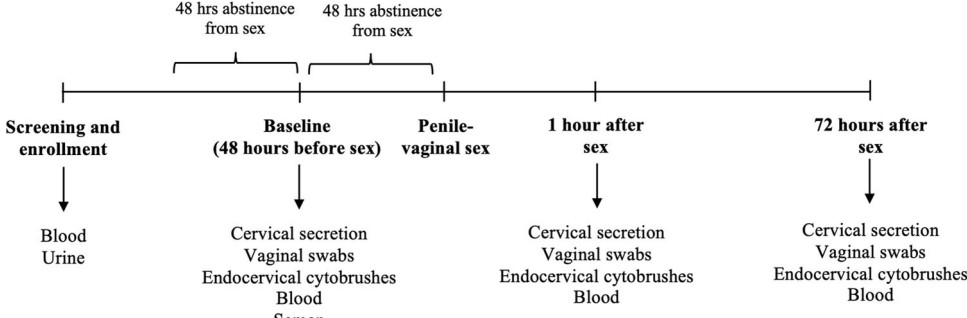

**Fig. 1 Overview of the study protocol.** Diagrammatic illustration of the study protocol and samples collected at each visit.

asked to abstain from sex 48 h before the baseline visit. At the baseline visit, participants completed a demographic/behavioural questionnaire and samples were collected. Couples were asked to abstain from sex for 48 h after the baseline visit before having one episode of penile-vaginal sex for the SECS study. Couples had either condomless sex or sex with a condom, and repeat sampling was performed after 1–2 h, and 72 h post-sex. Condom use during penile-vaginal sex was based on the preference of the couple. Women were advised to not wash their genital area after sex. Samples were collected in the following order: Softcup self-collection, study nurse vaginal swab collection, study nurse endo-cervical cytobrush collection, and blood collection. Cervico-vaginal secretions were self-collected using an Instead Softcup (Evofem, San Diego, CA) inserted for 1 min and were used for cytokine and Prostate Specific Antigen (PSA) analysis. Two vaginal swabs, two endocervical cytobrushes and blood were collected by the study nurse. Each cytobrush was gently inserted into the cervical os, rotated through 360°, placed into R10 medium (RPMI 1640 with 10% heat-inactivated FBS [Sigma-Aldrich, Carlsbad, CA], 100 mg/ml streptomycin, 100 U/ml penicillin, and 13 GlutaMAX-1 [Life Technologies, Grand Island, NY] media) at 4 °C and transported to the laboratory within 30 min of collection. The two cytobrushes were processed together, and the combined cells filtered through a 100-μm filter, washed, and divided into two equal aliquots for staining.

Peripheral blood mononuclear cells (PBMCs) were isolated by Ficoll-Hypaque density centrifugation at 1500 rpm for 30 min, counted, and washed in R10 medium. One aliquot of one million PBMCs was used for staining of T cell subsets.

Men provided semen samples at baseline approximately 1 h before attending clinic. Semen was collected by masturbation into a sterile container with 10 ml RMPI.

**STI and BV diagnosis**. As previously described[42], a vaginal swab was smeared onto a glass slide, air-dried and Gram's stained to diagnose bacterial vaginosis (BV) using Nugent criteria and to screen for vaginal yeast. Testing for GC and CT were done on first-void urine at Mount Sinai hospital by nucleic acid amplification test (NAAT; ProbeTech Assay, BD, Sparks, MD). Testing for HIV1/2 and syphilis were performed by chemiluminescent microparticle immunoassay (CMIA) (ARCHITECT System, Abbott GmbH & Co. KG).

**Immune cell phenotyping**. Endocervical cells, PBMCs and whole blood were surface stained with two panels of Abs to characterize various T cell subsets and neutrophils/APC populations. The T cell panel consisted of CD45RA-FITC (BioLegend, 1 μl/test), CD8- Percp cy5.5 (eBioscience, 1 μl/test), β7-APC (BD Biosciences, 2.5 μl/test), CD127-APCef780 (eBioscience, 2.5 μl/test), CD25-BV421(BD Biosciences, 1 μl/test ), CD4-BV650 (BD Biosciences, 1 μl/test), CCR6-BV711 (BD Biosciences, 2.5 μl/test), CD3-BV785 (BD Biosciences, 1 μl/test), α4-PE (BD Biosciences, 2.5 μl/test), CCR5-PE-CF549 (BD Biosciences, 2.5 μl/test), CCR7-Pe-cy7 (BD Biosciences, 2.5 μl/test), HLA-DR-BUV395 (BD Biosciences, 2.5 μl/test), CD69-BUV737 (BD Biosciences, 5 μl/test) and Live/Dead Aqua (Invitrogen, 1 μl/test). The neu-trophils/APCs panel consisted of CD14-FITC (BioLegend, 2.5 μl/test), CD66b-percp-cy5.5 (BioLegend, 2.5 μl/test), BDCA-2-APC (BioLegend, 5 μl/test), CD45-APC-fire (BioLegend, 1 μl/test), CD16-BV421 (BD Biosciences, 1 μl/test), CD83-BV650 (BD Biosciences, 2.5 μl/test), CD11c-BV785 (BD Biosciences, 1 μl/test), CD15-Pe (BD Biosciences, 5 μl/test), CD123- PECF549 (BD Biosciences, 5 μl/test), CD86-Pe-cy7 (BD Biosciences, 1 μl/test), HLA-DR- BUV395 (BD Biosciences, 2.5 μl/test), CD3/CD19- BUV-737 (BD Biosciences, 2.5 μl/test) and Live/ Dead Aqua (Invitrogen,

1 μl/test). Cells were enumerated using a BD LSR Fortessa X20 flow cytometer (BD Systems) and analyzed with FlowJo 10.4.1 software (TreeStar, Ashland, OR) by the same researcher for consistency. For the T cell panel, Fluorescence Minus One (FMO) control, and for the APC panel isotype controls were used for gating.

For each cytobrush sample, all isolated endocervical cells were run through the cytometer, allowing for the endocervical immune cell populations to be quantified as both a proportion (%) of and a total number of cells/cytobrush. In the T cell panel, samples with a CD4+ cell count lower than 10 were excluded from the proportion analysis. In the APC panel, the same exclusion strategy was applied to samples in regard to CD14+ or CD14- cell count.

**Cytokine analysis**. Cervico-vaginal secretions (CVS) collected by Softcup as described above were diluted 10-fold using sterile PBS and spun at 1730 g, 4 °C for 10 min. Subsequently, the super-natant was frozen at −80 °C for cytokine analysis. The volume of each semen sample plus collection media (RPMI) was recorded to calculate the dilution factor. Semen samples were spun down at 1000 g for 10 min at 4 °C and the supernatant was frozen at −80 °C. The levels of cytokines IL-1α, IP-10, IL-8, MIP-3α, MIP-1β, IL-17a, IFN-α2a, IL-6, MIG, sE-cad and MMP9 were mea-sured in duplicate by multiplex immune assay (Meso Scale Dis-covery, Rockville, MD) as previously described[42]. The samples were plated at 25 μl per well. A standard curve was used to determine the lower and upper limit of detection and the con-centration of each analyte (pg/ml).

Any sample above the upper limit level of detection was diluted and the multiplex immunoassay repeated for that sample. The highest lower limit of detection (LLOD) value across runs was selected as the LLOD for each analyte. The LLODs were as follow: IFNα2a = 0.28 pg/ml; IL-17 = 1.20 pg/ml; MIP-3α = 4.27 pg/ml; IL-6 = 0.25 pg/ml; IL-1α = 20.6 pg/ml; IL-8 = 0.10 pg/ml; MIG = 0.086 pg/ml; IP-10 = 0.99 pg/ml; MIP-1β = 13.9 pg/m; sE-cad: 53.9 pg/ml; MMP-9: 0.35 pg/ml. Samples that were below the limit of detection (LLOD) were given the LLOD value. Samples that were above the LLOD value but below LLOD + 30% with a high CV were not repeated and the LLOD value was given to those samples. Samples that were above the LLOD with a CV repeatedly higher than 30 were excluded from analysis.

All samples were run by a researcher blinded to the status of participants. Samples provided by each couple at all study visits were assessed on the same plate to account for the between plate variability.

**Statistics and reproducibility**. The co-primary endpoints for the SECS study were the vaginal levels of the chemoattractant che-mokine IP-10 and the pro-inflammatory cytokine IL-1α imme-diately (1 h) after sex. While analysis of other immune factors was felt to be important for this pilot study, these results were not corrected for multiple comparisons and should be considered exploratory. The sample size calculation was based on a small pilot study that assayed inflammatory cytokines in genital secretions. Based on the mean and SD of vaginal IL-1 levels, in order to detect a 40% increase after penile vaginal sex with alpha = 0.05 and power = 80% would require a sample size of $n = 20$ couples; in order to permit subanalysis based on male partner circumcision status, we doubled this to $n = 40$ couples.

For group comparisons, non-parametric statistical analyses were performed to reduce the effect of outliers. Comparisons of baseline immune factors (cells and cytokines) were performed using a Mann-Witney U test. Changes in the proportion/number of immune factors (cells and cytokines) between time points were

**Table 1 Participant characteristics (n = 36).**

| Characteristic | Total (n = 36) |
|---|---|
| Age | 22 (18–33) |
| Ethnicity n. (%) | |
| Asian (East or South Asian) | 15 (41.7) |
| White | 12 (33.3) |
| African-Caribbean and Black (ACB) | 2 (5.6) |
| Middle Eastern | 1 (2.8) |
| Latin American | 1 (2.8) |
| Mixed | 5 (13.9) |
| Circumcision status in the male partner (n) | |
| Circumcised | 17 (47.2) |
| Uncircumcised | 19 (52.8) |
| Contraceptive method | |
| Oral hormonal Contraceptive | 10 (27.8) |
| Condom | 9 (25) |
| Hormonal IUD | 4 (11.1) |
| IUD (Copper) | 3 (8.3) |
| Condom and hormonal contraceptive | 3 (8.3) |
| NuvaRing | 2 (5.6) |
| None | 5 (13.9) |
| Time since last sex (days) | 4 (2–23) |
| Episode of sex (last month) | 6 (1–25) |
| Relationship length (months) | 18 (1–96) |
| Type of previous sex % | |
| Ever vaginal | 100% |
| Ever Oral | 97.2% |
| Ever anal | 25% |
| Self-reported STI (ever) % | 8.3% |
| Douching % | 11.1% |
| Bacterial vaginosis % (defined by Nugent Score) | 13.5% |

analysed using a Wilcoxon Signed Ranked test. The change in the levels of IP-10 and IL-1α relative to baseline was analysed using a Wilcoxon Signed Ranked test, using a $p$-value threshold of <0.05 for significance.

To test whether differences in the within-person CVS concentration of cytokines post condomless sex could be explained by baseline concentrations of cytokines in semen, we performed a repeated measures ANOVA of the change in the CVS cytokines (IP-10 and MIG) at two time points post condomless sex (1 and 72 h) and compared these results to the same model but including the baseline concentration of the cytokines in semen as covariates (ANCOVA). All models were examined to test whether they met Mauchy's test of sphericity. In case the sphericity assumption was violated, the Huynh-Feldt estimate was reported.

To investigate the impact of BV status on changes in cytokine levels post-condomless sex, we performed a repeated measures ANOVA of the change in levels of cytokines in CVS at three time points (baseline, 1 h and 72 h) and included BV status as a between subject factors. For this analysis, we focused on the cytokines that had exhibited significantly different levels at baseline between BV+ vs BV- women with the Mann-Witney U test.

Data analysis were performed using IBM SPSS v.24 and graphs were prepared by GraphPad Prism v.7.

**Reporting Summary**. Further information on research design is available in the Nature Research Reporting Summary linked to this article.

## Results

**Participant enrolment and characteristics**. Consenting couples were invited to participate in the SECS protocol through the Women's Health in Women's Hands Community Health Center (WHIWH) in Toronto, Canada. Eligible, STI-free couples ($N = 39$ couples) were requested to abstain from sex for 48 h prior to the baseline genital sampling visit, and then again for 48 h prior to engaging in penile-vaginal sex and undergoing longitudinal post-coital sampling at 1 h and 72 h (Fig. 1). Condom use during penile-vaginal sex was based on the preference of the couple. Abstinence was verified by testing vaginal secretions for the presence of Prostate Specific Antigen (PSA) at all study visits. "Condomless sex" was defined based on couples' self-report and the detection of PSA in vaginal secretions 1 h post-sex. Three couples with a positive PSA result at the baseline (pre-sex) visit were excluded from further analysis. Therefore, the final sample set consisted of 36 couples; with 29 couples in the condomless and 8 couples in the condom-protected penile-vaginal sex group (one couple participated twice, once in each group). In addition, three couples had a positive PSA result at the 72 h post-sex visit (indicating condomless sex during the interim) and a fourth couple missed this visit; all four were excluded from the 72 h analysis. Full demographic data are shown in Table 1. The median age of female participants was 22 years (range, 18–33 years) and most (61%) reported contraceptive use other than condoms. Of the male partners, 17 were circumcised and 19 were uncircumcised. The median relationship duration prior to study participation was 18 months (range, 1–96 months).

The median self-reported duration of sexual abstinence prior to the baseline sampling visit was 4 days (range, 2–23 days), and from the baseline visit until having penile-vaginal sex for study purposes was 44 h (range, 38.5–78 h). The median time from completion of penile-vaginal sex to the first post-sex sampling visit was 75 min (range 20–120 min; Fig. 1), and to the second visit was 73 h (range, 48.5–103 h). A sampling visit was also scheduled at 8 h post-sex, but a validation study demonstrated that cytobrush sampling itself induced immune changes during this period[43], and so these samples were not included in the immune analysis.

**Comparison of baseline semen and cervico-vaginal cytokine levels**. Cytokine levels in semen would be expected to have an important impact on cervico-vaginal cytokine levels after condomless penile-vaginal sex. Therefore, the baseline level of each cytokine in our panel was compared between semen and cervico-vaginal secretions (CVS) using Mann-Whitney test. Levels of IL-1α, IL-17, IL-6, IL-8, MMP9 were enriched in CVS (all $p < 0.002$), levels of MIP1β- and MIP-3α were similar in CVS and semen (both $p > 0.18$), and levels of IP-10, MIG, sE-cad and IFNα2a were significantly enriched in semen (all $p < 0.001$; see Fig. 2).

**Impact of penile-vaginal sex on cervicovaginal cytokines**. Higher cervicovaginal levels of IP-10 and IL-1α in the FGT predicted subsequent HIV acquisition in a cohort of South African women[6], and were therefore pre-defined as our co-primary endpoints. To examine the impact of penile-vaginal sex on cervicovaginal cytokine/chemokine levels, the change from baseline (pre-sex) in undiluted CVS cytokine concentration was calculated for the 1 h and 72 h post-sex visits, stratified by condom use. Vaginal levels of IP-10 increased immediately after sex, regardless of condom use (median difference$_{condomless}$ = +2,213.24 pg/ml, $p = 0.005$; median difference$_{condom}$ = +1,473.69 pg/ml, $p = 0.012$, respectively; Fig. 3a, b). Cervicovaginal levels of IL-1α also increased rapidly in both groups, although the increase only reached significance in the condom-protected group (median difference$_{condomless}$ = +12,581.96 pg/ml, $p = 0.157$; median

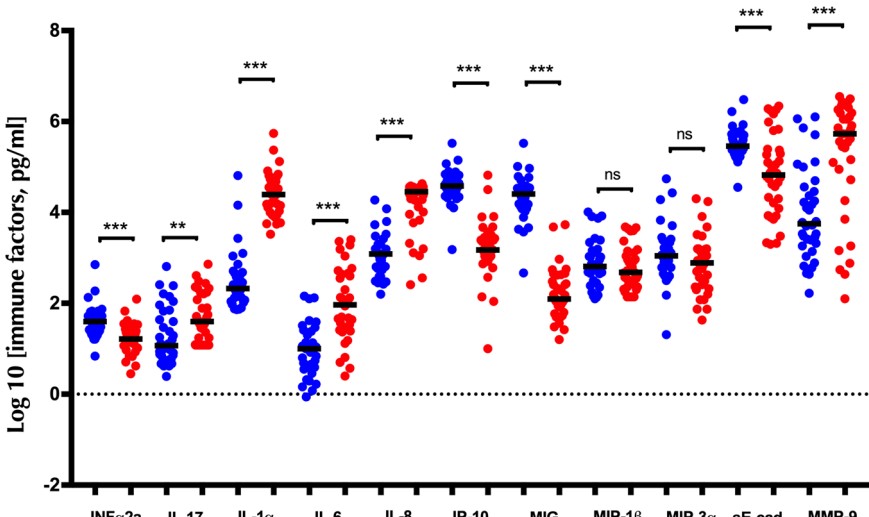

**Fig. 2 Baseline levels of cytokines in semen and cervico-vaginal secretions (CVS).** Baseline levels of cytokines in semen (blue dots) and cervico-vaginal secretions (red dots). Statistical comparisons were performed using Mann-Whitney test. $*p < 0.05$, $**p < 0.01$, $***p < 0.001$.

difference$_{condom}$ = +12,991.02 pg/ml, $p = 0.012$, respectively; Fig. 3c, d). Levels of both cytokines then fell to baseline by 72 h, regardless of condom use (both $p > 0.15$; Fig. 3a–d).

Secondary endpoints included the levels of several other chemokines (IL-8, MIG, MIP-1β and MIP-3α) and pro-inflammatory cytokines (IL-6, IL-17). Levels of MIG and MIP-1β also increased significantly immediately after sex, regardless of condom use, and returned to baseline by 72 h (Fig. 3e–h). No significant changes were seen after sex in the levels of MIP-3α, IL-6 or IL-17 (Fig. 3e–h). There was a transient increase in IFNα2a after condomless sex ($p = 0.004$ at 1 h and $p = 0.949$ at 72 h, respectively; Fig. 3e) and a delayed increase in IL-8 at 72 h after condom-protected sex ($p = 0.028$, Fig. 3h).

As biomarkers of epithelial disruption after penile-vaginal sex we quantified levels of sE-cad and the neutrophil protease MMP9, with elevated levels indicating reduced barrier integrity. sE-cad increased immediately after sex, regardless of condom use (both $p < 0.02$; Fig. 3e, f), and similar trends were seen for MMP9 ($p = 0.006$ and $p = 0.069$, respectively; Fig. 3e, f). Both parameters returned to baseline within 72 h of penile-vaginal sex (both $p > 0.17$; Fig. 3g, h).

In summary, cervicovaginal proinflammatory cytokines/chemokines and biomarkers of epithelial disruption increased immediately (1–2 h) after penile-vaginal sex, regardless of condom use, and returned to baseline levels within 72 h.

**Impact of semen parameters on cervico-vaginal cytokine changes after condomless sex.** Condom-protected sex led to a rapid increase in several cytokines/chemokines, clearly indicating that penile-vaginal sex itself induced immune changes in the female genital tract, independent of semen exposure. However, since semen was enriched for several cytokines relative to CVS (particularly MIG and IP-10; see above and Fig. 2), we hypothesized that semen exposure might have influenced the vaginal cytokine changes observed after condomless sex. This was not deemed to be plausible for sE-cadherin, since the post-sex increase seen in the genital tract far exceeded sE-cad levels present in the semen of their male partners (median$_{CVS\ 1hr}$ = 1,132,390.43 pg/ml;

median$_{semen}$ = 286,592.44 pg/ml, respectively; $p = 0.001$; Supplementary Fig. 1). However, since semen levels of IP-10 and MIG were much higher than those seen in CVS immediately after sex (Supplementary Fig. 1), it was felt that semen could be an important contributor to these changes. Therefore, we performed a repeated-measures ANOVA on the change in log10 transformed concentration of CVS cytokines (IP-10 and MIG) at two time points in the post condomless sex (1 and 72 h) group by comparing models both without (ANOVA) and with baseline semen cytokine concentration as a covariate (ANCOVA), since the latter will control for exposure to IP-10 and MIG in semen. In the non-covariate model, there was a significant within person linear decrease in CVS cytokines (IP-10 and MIG) over time ($F_{(2, 50)} = 9.02$, $p < 0.001$; $F_{(1.7, 42.6)} = 22.98$, $p < 0.001$, respectively). However, after adjusting for baseline IP-10 and MIG in semen (ANCOVA), the within subject effects were not significant ($F_{(2, 48)} = 0.35$, $p = 0.703$; $F_{(1.8, 45.4)} = 1.77$, $p = 0.182$, respectively), indicating that the higher concentrations of IP-10 and MIG in semen could explain the cervicovaginal changes seen after condomless sex.

**Impact of bacterial vaginosis on cervico-vaginal cytokine changes after condomless sex.** Bacterial vaginosis (BV) is associated with elevated levels of pro-inflammatory vaginal cytokines[44,45], and thus we investigated the impact of BV status on baseline CVS cytokine levels and on CVS cytokine changes after condomless sex. Women with BV at baseline, defined as a Nugent score ≥7 ($n = 5$), had significantly higher vaginal levels of IL-1α, sE-cad and MMP9 compared to BV negative women (Supplementary Fig. 2; IL-1α median$_{BV+}$ = 132,805.22 pg/ml versus median$_{BV-}$ = 23,507.52 pg/ml, $p = 0.004$; sE-cad median$_{BV+}$ = 1,809,061.52 pg/ml versus median$_{BV-}$ = 58,387.16 pg/ml, $p = 0.001$; MMP9 median$_{BV+}$ = 2,259,137.51 pg/ml versus median$_{BV-}$ = 314,963.46 pg/ml, $p = 0.006$; respectively). In addition, they had lower vaginal levels of IP-10 (median$_{BV+}$ = 138.06 pg/ml versus median$_{BV-}$ = 2,294.61 pg/ml, $p = 0.001$).

Given these immune differences, we next assessed whether BV status impacted the effect of condomless sex on cytokine levels, with a focus on those cytokines that were significantly different at

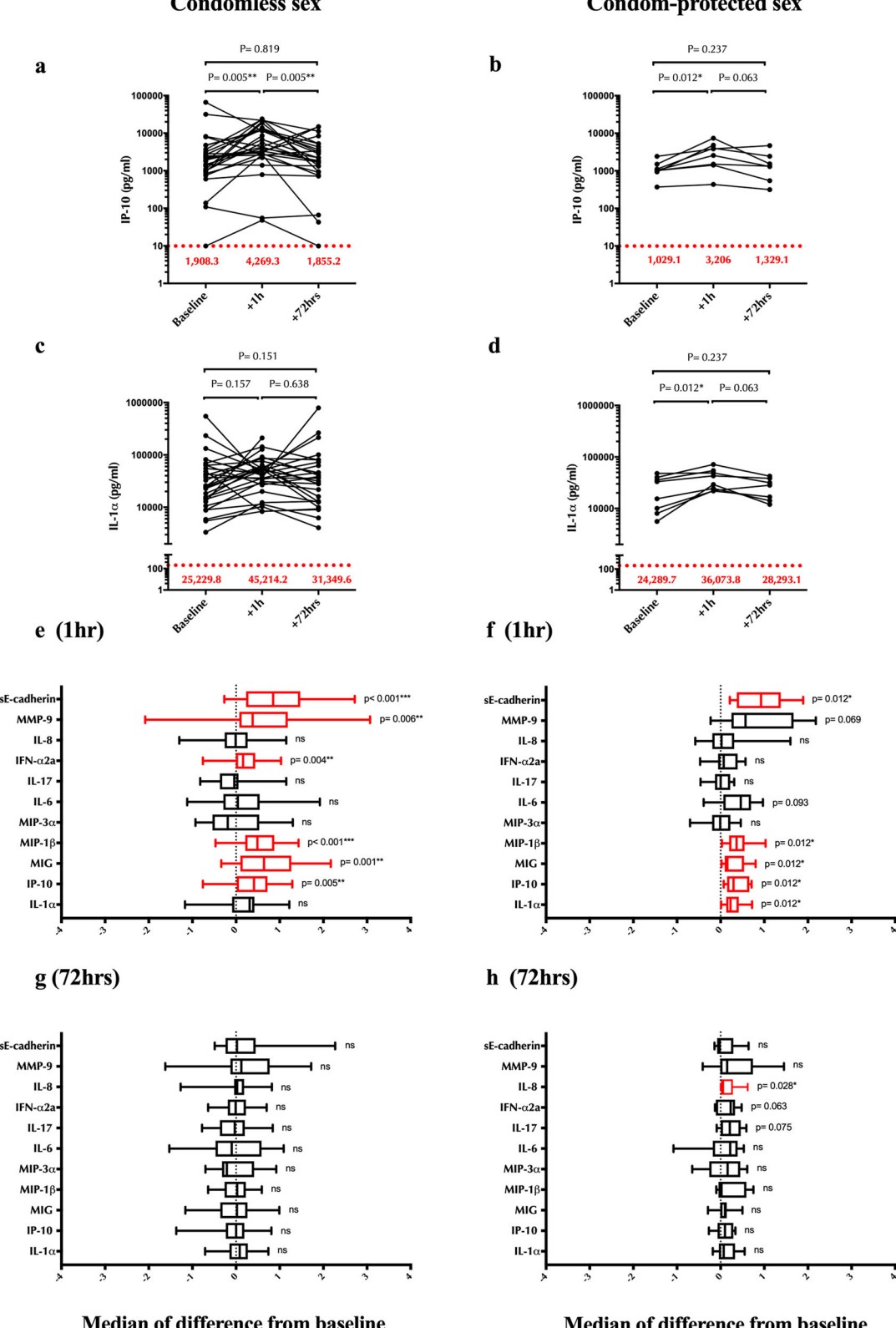

**Condomless sex** — **Condom-protected sex**

**e (1hr)** — **f (1hr)**

**g (72hrs)** — **h (72hrs)**

**Median of difference from baseline** — **Median of difference from baseline**

baseline in women with BV. Specifically, we performed a repeated measures ANOVA examining the change in log10 transformed levels of IL-1α, sE-cad, IP-10 and MMP9 in CVS at three time points (baseline, 1 h and 72 h) in the post condomless sex group and included BV status as a between-subject factor. This analysis demonstrated significant interactions between BV status and changes in cervicovaginal IL-1α and sE-cad after condomless sex ($F_{(2,52)} = 13.45$, $p < 0.001$; $F_{(2,52)} = 7.96$, $p = 0.001$, respectively), meaning that the impact of condomless sex on these cytokines was different in women with and without BV. Stratifying the data based on BV status revealed that while the levels of IL-1α and sE-cad increased 1 h post condomless sex in

**Fig. 3 Impact of penile-vaginal sex on cervico-vaginal soluble immune factors.** Concentration of IP-10 (pg/ml) at baseline and 1 h, 72 h after (**a**) condomless sex ($N = 29$), (**b**) condom-protected sex ($N = 8$). Concentration of IL-1α (pg/ml) at baseline and 1 h and 72 h after (**c**) condomless sex ($N = 29$), (**d**) condom-protected sex ($N = 8$). The red dotted line shows the lower level of detection (LLODs). The red numbers illustrate the median cytokine concentration at each visit. See methods for detailed information on LLOD values for each cytokine. Statistical comparisons were performed using Wilcoxon Signed Ranked test. Summary of the median difference and interquartile range of undiluted cervico-vaginal cytokines concentrations from baseline to 1 h post (**e**) condomless sex and (**f**) condom-protected sex. Summary of the median difference and interquartile range of undiluted cervico-vaginal cytokines concentrations from baseline 72 h post (**g**) condomless sex and (**h**) condom-protected sex. Cytokine values were log10 transformed only for the presentation purpose, and statistical comparisons were performed using Wilcoxon Signed Ranked test. $*p < 0.05$, $**p < 0.01$, $***p < 0.001$. The box plots represent the median difference from baseline, with interquartile and minimum to maximum ranges.

BV negative women, these cytokines actually decreased in BV + women. No significant interactions were observed for IP-10 and MMP-9 (both $p > 0.15$).

**Male partner circumcision status and cervicovaginal immune changes after sex.** We next examined the impact of male partner circumcision status on vaginal immunology after condomless sex. Demographic characteristics were similar between groups (Supplementary Table 1). Prior to sex, the circumcision status of the male partner was not associated with immune differences in his semen or in the CVS of the female partner (all $p > 0.08$; Supplementary Tables 2, 3). In the same cohort, we have shown previously that levels of coronal sulcus cytokines were also comparable between uncircumcised and circumcised men[42]. However, immediately after condomless sex there were a pronounced increase in vaginal IL-1α among women with uncircumcised male partners (median difference $_{uncircumcised} = +18,519.72$ pg/ml, $p = 0.020$; Fig. 4a) that was not apparent in women with a circumcised male partner (median difference $_{circumcised} = +2,905.47$ pg/ml, $p = 0.972$; Fig. 4a). Likewise, an immediate increase in vaginal sE-cad levels was only apparent in women with uncircumcised male partners (median difference $_{uncircumcised} = +560,246.07$ pg/ml, $p = 0.002$; median difference $_{circumcised} = +333,072.54$ pg/ml, $p = 0.075$, respectively; Fig. 4d), although similar trends were apparent in both groups. These increases in vaginal IL-1α and sE-cad had resolved completely by 72 h. Interestingly, vaginal levels of MIG and IFNα2a only increased significantly in women with circumcised male partners (for MIG: median difference $_{uncircumcised} = +126.4$ pg/ml, $p = 0.088$ versus median difference $_{circumcised} = +2,353.19$ pg/ml, $p = 0.004$; for IFNα2a: median difference $_{uncircumcised} = +8.04$ pg/ml, $p = 0.121$ versus median difference $_{circumcised} = +20.53$ pg/ml, $p = 0.012$; Fig. 4c, j, respectively), although again similar trends were apparent in each group. The impact of sex on levels of soluble immune factors IP-10, IL-8, IL-6, IL-17, MIP-1β, MIP-3α, and MMP9 did not differ based on male partner circumcision status (Fig. 4).

**Impact of penile-vaginal sex on highly susceptible HIV target cells in the cervix.** Alterations in genital pro-inflammatory cytokines/chemokines may recruit HIV-susceptible cells to the mucosa, with Th17 cells being key HIV targets[8,11–14]. Therefore, we next assessed the impact of penile-vaginal sex on the number and proportion of endocervical cytobrush-derived Th17 cells (gating strategy, Fig. 5a). The proportion of endocervical Th17 cells increased significantly one hour after condomless sex (median difference = $+5.1\%$, $p = 0.007$; Fig. 5b), although there was no increase in the absolute number of CD4 + T cells or Th17 cells per cytobrush (both $p > 0.5$). This increase in the proportion of cervical Th17 cells resolved within 72 h ($p = 0.282$; Fig. 5b), and was not observed after condom-protected sex (both $p > 0.063$; Fig. 5b).

We next investigated the impact of sex on other CD4 + T cell subsets that may constitute preferential HIV targets, including activated CD4 + T cells (HLA-DR + ; see Supplementary Fig. 3 for

gating strategy) and memory subsets (effector, $T_{EM}$ and central, $T_{CM;}$ see Supplementary Fig. 3 for gating strategy). Again, sex did not change the absolute number of cells per cytobrush, but the proportion of activated T cells and central memory cells (Tcm) (CD45RA-CCR7 + ) increased immediately after condomless sex (both $p < 0.03$; Fig. 6a) and remained elevated at 72 h (both $p < 0.006$; Fig. 6c). Similar effects were generally observed after condom-protected sex despite lower participant numbers: the proportion of activated cervical CD4 + cells was significantly elevated at 72 h, and of $T_{CM}$ immediately after sex (Fig. 6b, d, respectively). There were no significant differences in the proportion or number of CCR5 + cells after condomless sex (all $p > 0.2$; Fig. 6a, c); However, the proportion of CCR5 + CD4 + T cells increased significantly 72 h after condom-protected sex ($p = 0.028$; Fig. 6d).

**Effects of penile-vaginal sex on cervical dendritic cell subsets.** Genital monocyte-derived DCs (CD14 + DCs) may be an important HIV target[17], and so we next explored the activation (CD86 expression) and maturation status (CD83 expression) of monocyte-derived DCs (CD14 + DCs) (gating strategy; Supplementary Fig. 4). The percentage of both activated and mature monocyte-derived DCs (CD14 + DCs) increased immediately after condomless sex (median difference = $+2.7 \%$, $p = 0.046$; median difference = $+6.4\%$, $p = 0.014$, respectively; Fig. 6a); while monocyte-derived DCs (CD14 + DC) activation resolved by 72 h, the maturation markers remained persistently elevated (median difference = $-1.1\%$, $p = 0.809$; median difference = $+9.4\%$, $p = 0.012$, respectively; Fig. 6c). A similar trend, albeit non-significant, was apparent after condom-protected sex (all $p > 0.06$; Fig. 6b, d).

Migratory CD14 negative DCs may play a role in HIV dissemination[17], and again the percentage of activated and mature CD14 negative DCs increased early after condomless sex (median difference = $+4.8\%$, $p = 0.004$; median difference = $+8.4\%$, $p = 0.004$, respectively; Fig. 6a), and returned to baseline levels by 72 h (both $p > 0.5$; Fig. 6c). Transient increases in activation were also seen after condom-protected sex (median difference $_{1hr} = +18.2\%$, $p = 0.043$; median difference$_{72hr} = +13.8\%$, $p = 0.345$; Fig. 6b, d).

**Cervical cell changes and male partner circumcision status.** In general, penile immunology and microbiome are very different in circumcised and uncircumcised men[31,46]. Thus, we next compared cervical cell changes after condomless penile-vaginal sex in women with a circumcised ($n = 13$) vs uncircumcised ($n = 16$) male partner. After condomless sex, the proportion of Th17 cells only increased significantly in women with uncircumcised male partners (median difference $_{uncircumcised} = +5.8\%$, $p = 0.004$ vs. median difference $_{circumcised} = +2.4\%$ and $p = 0.422$; Fig. 7a). Although cervical changes in activated CD4 + T cells and Tcm after condomless sex were similar (Fig. 7b, c, respectively), dendritic cell changes were also highly

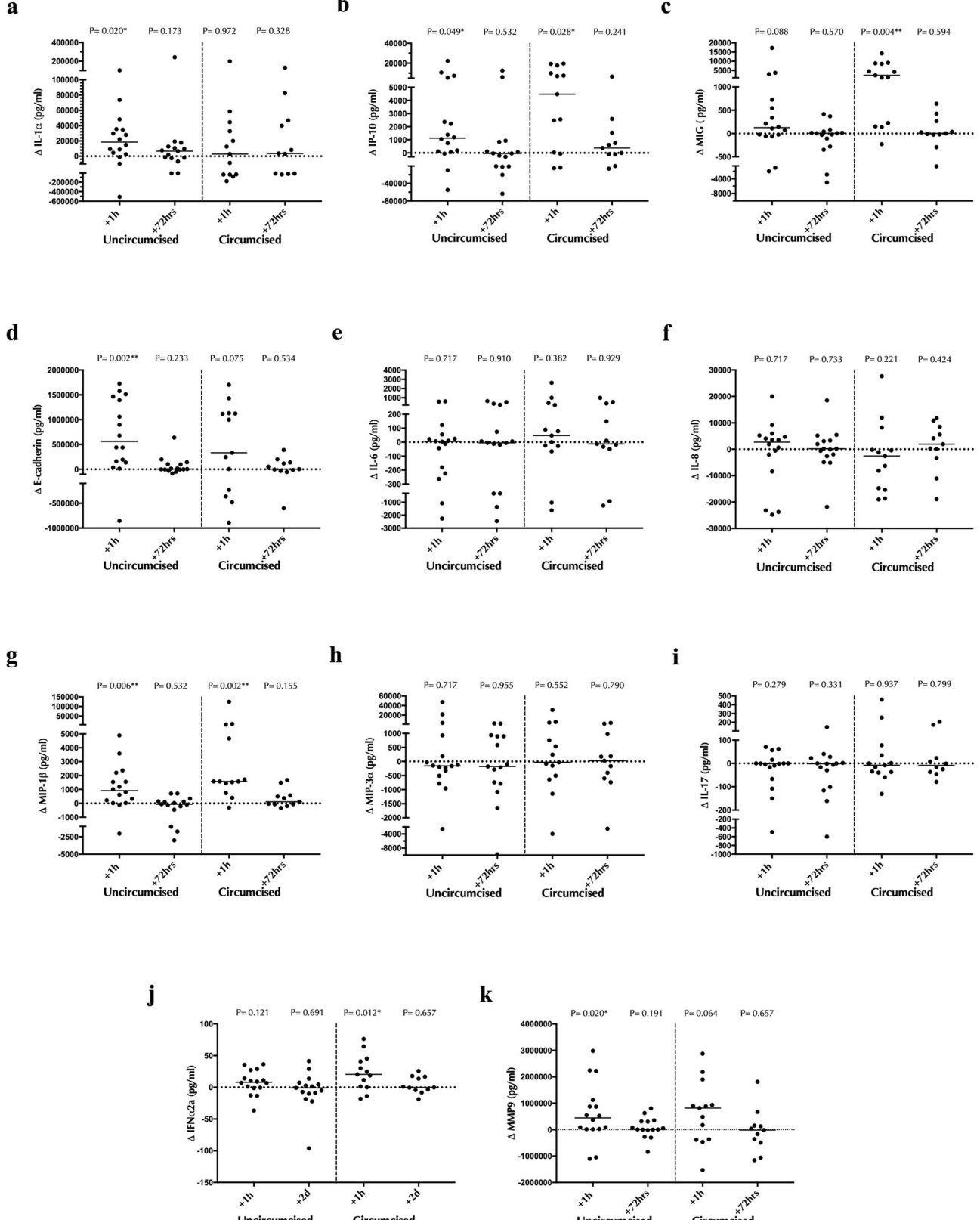

**Fig. 4 Impact of penile circumcision status on cervico-vaginal soluble immune factor changes after condomless sex.** Participants were divided into uncircumcised ($N = 16$), and circumcised ($N = 13$) groups based on circumcision status of the male partners. The change (Δ) in the concentration of cytokines 1 h and 72 h after condomless sex relative to baseline was defined in each group and is shown on the y-axis. **a** Δ IL-1α (pg/ml). **b** Δ IP-10 (pg/ml). **c** Δ MIG (pg/ml). **d** Δ sE-cadherin (pg/ml). **e** Δ IL-6 (pg/ml). **f** Δ IL-8 (pg/ml). **g** Δ MIP-1β (pg/ml). **h** Δ MIP-3α. **i** Δ IL-17 (pg/ml). **j** Δ IFNα2a (pg/ml). **k** Δ MMP9 (pg/ml). Statistical comparisons were performed using Wilcoxon Signed Ranked test. *$p < 0.05$, **$p < 0.01$, ***$p < 0.001$.

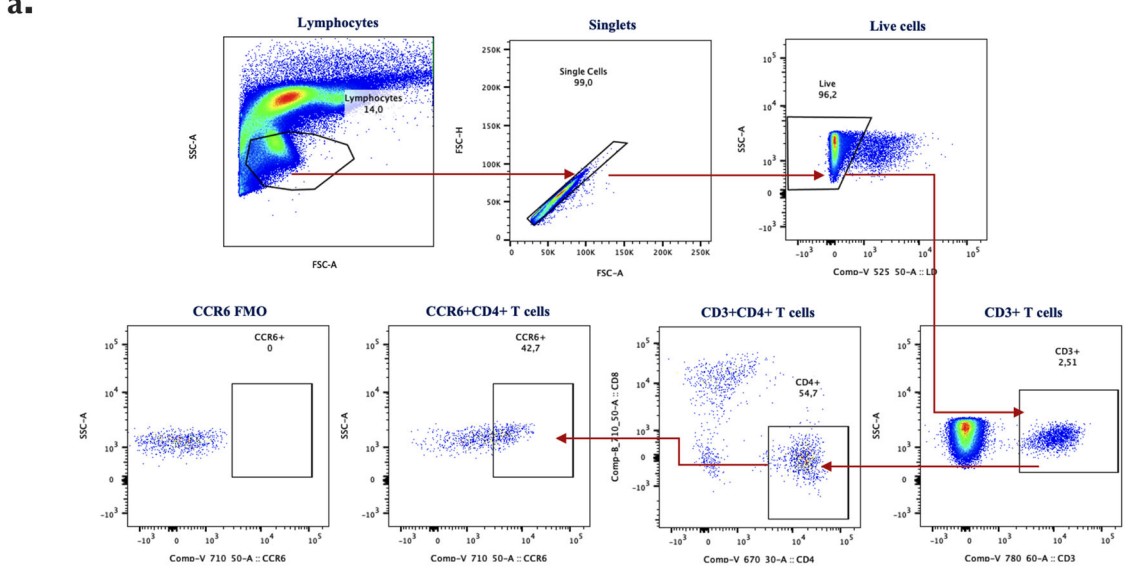

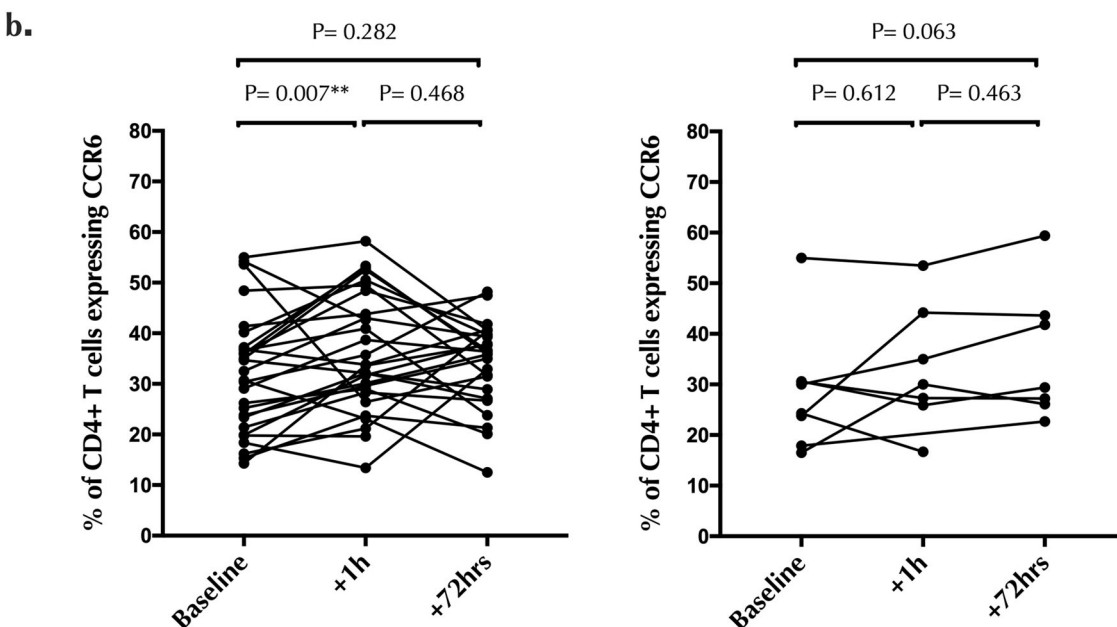

**Fig. 5 Endocervical Th17 changes post-sex. a** Gating strategy and representative plots for endocervical CD4 + T cells and Th17 cells. Cells were gated on lymphocytes, singlets, live, CD3 + cells, CD4 + cells, CCR6 + CD4 + T cells (Th17). **b** Endocervical Th17% changes 1 h and 72 h after condomless sex (N = 28) and sex with condom (N = 8). Statistical comparisons were performed using Wilcoxon Signed Ranked test. *p < 0.05, **p < 0.01, ***p < 0.001.

dependent on male circumcision status. Specifically, activation of monocyte-derived DCs (CD14 + DCs) and CD14- DCs increased early after condomless sex in women with uncircumcised male partners (median difference = +3.5%, $p = 0.028$, Fig. 8a; median difference = +7.8%, $p = 0.017$, Fig. 8c, respectively) but did not change in women with circumcised male partners (median difference = +0.25%, $p = 0.530$, Fig. 8a; median difference = +4.8%, $p = 0.083$, Fig. 8c, respectively). Likewise, the proportion of mature monocyte-derived DCs (CD14 + DCs) and CD14- DCs increased early in women with uncircumcised male partners (median difference = +19%, $p = 0.011$, Fig. 8b; median difference = +11%, $p = 0.004$, Fig. 8d, respectively), but not in those with circumcised male partners (median difference = +1.8%,

$p = 0.610$, Fig. 8b; median difference = +3.8%, $p = 0.328$, Fig. 8d, respectively).

## Discussion

Cervico-vaginal inflammation, regardless of its cause, is associated with an increased risk of HIV acquisition in prospective cohort studies[6,7,47]. Interestingly, genital inflammation is generally assessed in these studies after participants have abstained from sex for 48–72 h, despite the fact that the virus rapidly penetrates the genital epithelium after sex and is found in proximity to potential target cells within 4 h[48], with important early targets for virus infection including CD4 + T cell subsets and immature dendritic cells[11–13,17,49]. Therefore, our goal in the SECS study was to carefully assess the impact of a single episode

## Condomless sex (1 hour)

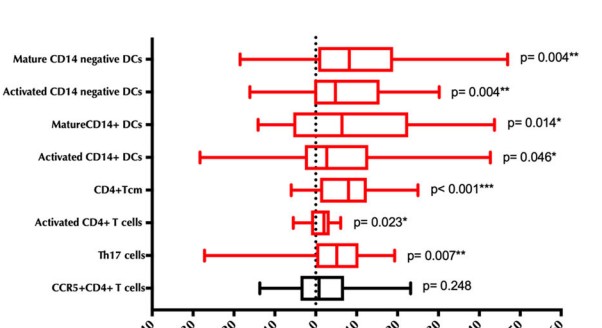

## Condom protected sex (1 hour)

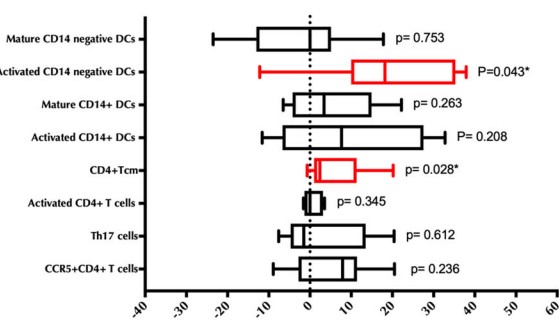

## Condomless sex (72 hours)

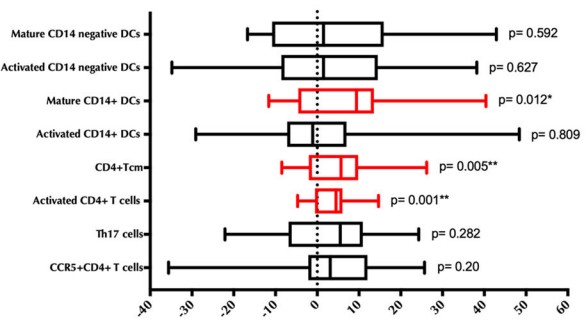

## Condom protected sex (72 hours)

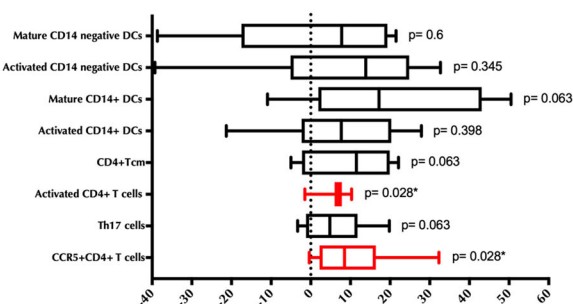

**Fig. 6 Impact of penile-vaginal sex on endocervical HIV target cells.** The change from baseline in the proportion of each cell subset was calculated at each time point. The median of percentage change and interquartile range are shown. Summary of the median of percentage difference and interquartile range of various CD4 + T cell subsets and Dendritic cells from baseline 1 h post (**a**) condomless sex ($N = 29$) and (**b**) condom-protected sex ($N = 8$). Summary of the median of percentage difference and interquartile range of various CD4 + T cell subsets and Dendritic cells from baseline 72 h post (**c**) condomless sex ($N = 26$) and (**d**) condom-protected sex ($N = 7$). Statistical comparisons were performed using Wilcoxon Signed Ranked test. *$p < 0.05$, **$p < 0.01$, ***$p < 0.001$. The box plots represent the median difference from baseline, with interquartile and minimum to maximum ranges.

of penile-vaginal sex on cervico-vaginal immune factors relevant to HIV acquisition in the short term (after 1 h) and medium term (after 72 h). Cervico-vaginal levels of several pro-inflammatory cytokines and chemokines previously linked to HIV acquisition increased immediately and transiently after both condomless and condom-protected sex, and there were also rapid increases in highly HIV-susceptible cervical cell populations, including Th17 cells, activated CD4 + T cells and activated DCs; in contrast to soluble immune factors, these cellular changes were more apparent after condomless sex, particularly in women with an uncircumcised male partner. These rapid immune alterations after sex may be critical in the short-term outcome of HIV exposure, and have clear importance for understanding the mucosal immunopathogenesis of HIV transmission.

An important mechanism by which cervico-vaginal inflammation enhances the risk of HIV transmission is the disruption of epithelial barrier integrity[8], which might also plausibly be affected by physical stresses during penile-vaginal sex. While the collection of biopsies was not part of our study protocol, epithelial barrier integrity was indirectly assessed through the release of sE-cad, since tissue-bound E-cadherin is an important component of the transcellular adherens junction in the endocervix, ectocervix and vagina[50]. sE-cad levels in genital secretions increased after penile-vaginal sex, even if a condom was used, suggesting that physical stresses during sex disrupt the epithelial barrier. A compromised epithelial barrier enhances virus access to HIV target cells in the mucosa and submucosa, and in vitro studies demonstrate that exposure of HIV target cells to inflammatory

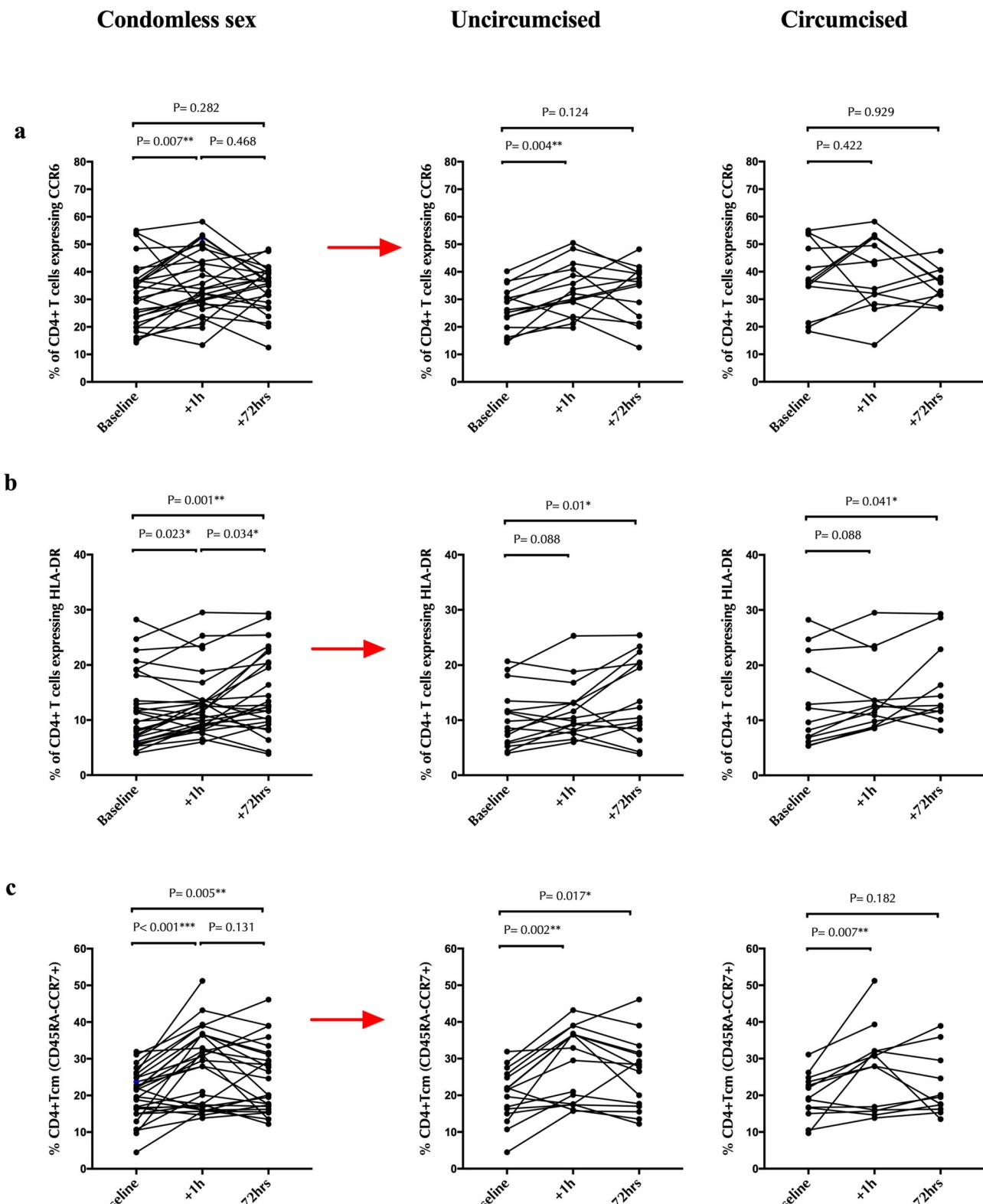

**Fig. 7 Impact of male partner circumcision status on changes in endocervical T cell subsets after penile-vaginal sex.** In each row, the graph on the left demonstrates changes in endocervical T cell subsets from baseline to 1 h and 72 h after condomless sex for all participants (N = 29). The central figure then shows data from the subset of couples with an uncircumcised partner, and the figure on the right from the subset of couples with a circumcised partner. Rows are: (**a**) %Th17, (**b**) % activated CD4 + T cells (HLA-DR + ), (**c**) % CD4 + central memory T cell (CD45RA-CCR7 + ). Statistical comparisons were performed using Wilcoxon Signed Ranked test. *p < 0.05, **p < 0.01, ***p < 0.001.

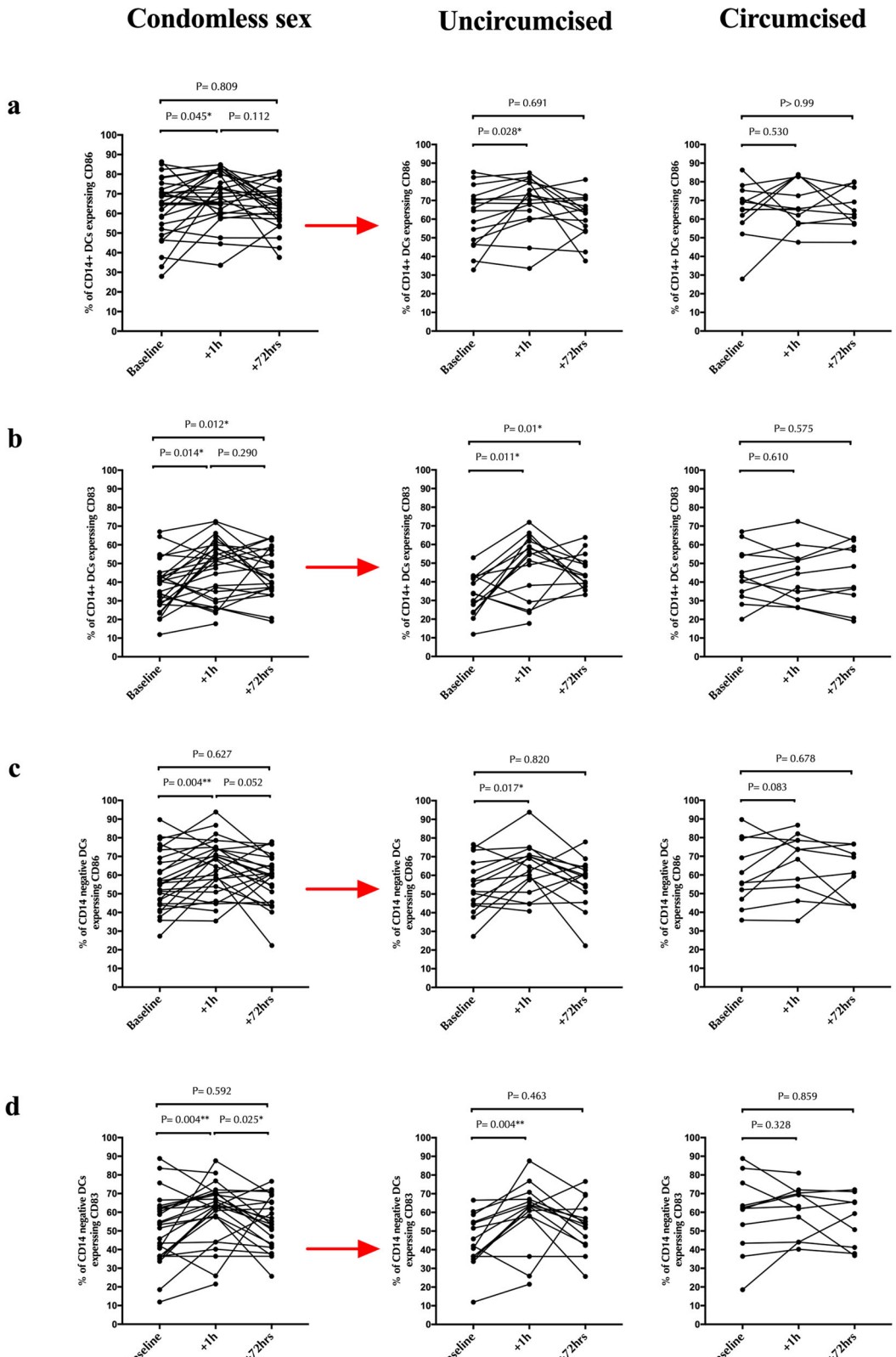

**Fig. 8 Impact of male partner circumcision status on changes in endocervical dendritic cells after penile-vaginal sex.** In each row, the graph on the left demonstrates changes in endocervical dendritic cells from baseline to 1 h and 72 h after condomless sex for all participants (N = 29). The central figure then shows data from the subset of couples with an uncircumcised partner, and the figure on the right from the subset of couples with a circumcised partner. Rows are: **a** % activated monocyte-derived DCs (CD86 + CD14 + DCs). **b** % mature monocyte-derived DCs (CD83 + CD14 + DCs). **c** % activated CD14- DC (CD86 + ). **d** % mature CD14- DCs (CD83 + ). Statistical comparisons were performed using Wilcoxon Signed Ranked test. * p < 0.05, ** p < 0.01, *** p < 0.001.

chemokines (including IP-10) also increases cellular HIV uptake[51]. While we found that both pro-inflammatory cytokine and sE-cad levels had returned to baseline within 72 h of sex, the rapid (within 4 h) viral penetration of the genital epithelium after sex that was demonstrated by Carias and colleagues suggests that this time frame may be critical for HIV acquisition[48]. Much higher levels of soluble E-cadherin and IL-1α were seen in the CVS of women with BV ($n = 5$), and the level of both cytokines actually decreased after condomless sex in these participants. While our sample size was not sufficient to explore the interaction between BV and post-sex immune changes any further, this may represent an important area for future research.

In addition to effects on epithelial integrity, genital inflammation may also enhance HIV susceptibility through the local recruitment of HIV target cells[8]. In the SECS study, the early chemokine increase after condomless sex coincided with an increase in the proportion of mucosal Th17 cells and activated CD4 + T cells, both of which are primary targets of HIV. Given that HIV must initially establish itself through local mucosal replication, prior to systemic dissemination[49], the increase that we observed in the proportion of preferential HIV target cells after condomless sex may be critical. After this local virus replication, dendritic cells (DCs) play an important role in the systemic dissemination of HIV[15,52], and so the early increase in genital DC activation and maturation that we observed after sex may also enhance HIV susceptibility. While we hypothesize that these DC alterations may relate to epithelial disruption, with DC exposure to soluble immune factors and/or microbiota components, this will require further investigation.

Penile circumcision reduces the risk of HIV acquisition in men by 60% or more[53–55], likely due to eradication of the subpreputial space, which is enriched in inflammatory bacteria and chemokines[56]. Although a small randomized trial did not demonstrate that circumcision of HIV-infected men reduced transmission to their female partners[57], circumcision of an HIV-uninfected man reduced the partner's incidence of severe bacterial vaginosis and several STIs[58]. In our SECS study approximately half the male partners were circumcised, and pre-planned stratification of our results based on circumcision status demonstrated that vaginal increases in the level of IL-1α, and in the proportion of cervical Th17 cells and activated/mature cervical DCs, were only observed after condomless sex in female partners of uncircumcised men. The mechanism for these differences is not known, and it is important to note that our sample size was relatively small and therefore these results need to be confirmed in future studies. However, the uncircumcised penis has a ten-fold higher total bacterial load and a much higher proportion of inflammatory anaerobic penile bacteria[46]. Since the density of some of these same bacteria in the vagina is a key determinant of inflammatory cytokine levels[59], this suggests that investigation of the genital microbiome as a basis for these immune differences after sex will be very interesting.

While these substantial effects of penile-vaginal sex on the immunology of the female genital tract are interesting, the SECS study does have some limitations. We assessed immune differences at 1 h and 72 h after sex but were not able to assess planned intermediate endpoints since our validation study demonstrated that cytobrush sampling itself induces immune changes that persist up to 72 h[43]. Therefore, future studies might limit sampling to swabs and/or SoftCup collection, which have less immune impact and would allow a more detailed assessment of immune dynamics after sex. Furthermore, our cohort was relatively small, and so we were not powered to perform numerous interesting subanalyses, such as the potential effects of participant use of hormonal contraceptives, hygiene practices, ethnic background, etc on the immune impacts of penile-vaginal sex.

Moreover, we defined Th17 cells based only on CD4 + T cell expression of CCR6, without surface markers such as CCR4 and/or CCR10 that would allow us to further define Th17 cell subsets. While these cells were demonstrated to be preferential early SIV targets in primate models[11], it is likely that a proportion of the cells we defined as Th17 may represent Th22 or Th1/Th17 cells.

In conclusion, the SECS study clearly demonstrates that receptive penile-vaginal sex induces rapid alterations in cervico-vaginal proinflammatory cytokines, endocervical cell populations and epithelial integrity as early as 1 h after sex. These changes had mostly resolved within 72 h, and would be expected to have important effects on host HIV susceptibility. Since there was heterogeneity in the immune effects of sex based on both host and partner characteristics, expanding these studies may be important to understand the mucosal immunopathogenesis of HIV transmission, and to better design female-focused HIV prevention strategies. It will also be important for cohort-based studies of mucosal immunology and HIV susceptibility to consider the immune changes induced by sex in their study design.

## Data availability

Source data analysed for the main figures can be accessed as Supplementary Data 1. All data generated in this study will be provided by the corresponding author on reasonable request.

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

## Acknowledgements

We acknowledge the time and cooperation of all study participants. We also thank the invaluable support we received from all staff at Women's Health in Women's Hands Community Centre who helped us with this project. Funding: Canadian Institutes of Health (CIHR; PJT-156123 and TMI-138656, R.K.).

## Author contributions

All authors contributed to the study. A.M. and R.K. conceptualized and designed the study. A.M., S.B., Y.C., A.F., E.T., S.H., W.T., and R.K. were involved in study execution. A.M. and S.V.G. performed data analysis. A.M. and R.K. were involved in the interpretation of the study.

## Competing interests

The authors declare no competing interests.
