## [Peer Review File · Communications Medicine]

Reviewers' comments:

Reviewer #1 (Remarks to the Author):

This is excellent study that adds critical information to our understanding of the dynamics of HIV transmission in cervicovaginal tissue. The authors first determine the cytokine profile of semen and cervicovaginal secretions and show that IL-1a, IL-17, IL-6, IL-8, MMP9 are enriched in cervicovaginal and IP-10, MIG, sE-cad and IFNa2a enriched in semen. They then show that vaginal levels of IP-10, IL-1a, MIG and MIP-1b increased immediately after sex, regardless of condom use. This is an important observation as IP-10 and IL-1a cytokines have been shown to predict HIV acquisition in women. Interestingly, other proinflammatory cytokines (IL-6) were not elevated post intercourse however evidence of epithelial barrier disruption was shown. As you might expect women with bacterial vaginosis had higher levels of baseline proinflammatory cytokines and lower IL-10. Interesting however these cytokines decreased in BV+ women post condomless sex and lower cervicovaginal IL-10 levels could be correlated to levels in semen and correlated with lack of circumcision. They next show that Th17 cells and dendritic cells are increased in cervix post intercourse which are known key HIV target cells as well as activated and central memory CD4 T cells. Finally, they show that changes are strongly correlated with a lack of circumcision which is a very important finding.

Line 56-57. Langerhans cells should be added to the list of HIV target cells. These are no longer considered a dendritic cell subset and should be defined as their class of mononuclear phagocyte. Also, there are several recent studies that show that DCs can transmit HIV CD4 T cells within 1-2 hours (e.g. PMID: 33846309, 31227717, 21738469) of virus capture and Th17 cells are present in genital tissues. Therefore, dendritic cells (DCs) probably transmit the virus to CD4 T cells within genital tissues as well as secondary lymphoid organs.

I think the baseline cytokine levels of cervicovaginal secretions vs semen are of key interest. The authors might like to consider moving this from supplemental data into the main body of data.

The flow cytometry gating strategy that defines Th17 T cells as CD3+CD4+CCR6+ cells is unconvincing. CCR6 is a tissue homing marker and this population is likely to also contain many other T cell subsets. RORgT should have been included to define Th17 cells. Although most cells are probably Th17 cells CXCR3 needs to be included to exclude Th1 and CCR10 to exclude Th22 cells. Furthermore, gating strategy for effector, TEM and TCM is not shown. It is important the authors present this.

The flow cytometry gating strategy (supp Figure 2 not 1 as stated) for cervical DCs however is excellent and relevant to the most recent literature that shows the CD14+ CD11c+ cells are DCs and CD14+CD11c- cells are macrophages (PMID 33846309). Nevertheless, the authors should refer to these as monocyte derived DCs as these are derived from blood monocytes as opposed to bona fide CD14- CD11c+ DCs derived from bone marrow. They should refer to the CD14+CD11c+ DCs as cDC2.

Reviewer #2 (Remarks to the Author):

In this study the authors investigated the effects of penile-vaginal sex on genital inflammation and

potential epithelial barrier alterations by measuring cytokine/chemokines, immune cell populations and soluble E-cadherin in genital samples before and after sex. The study addresses an important gap in knowledge and the findings will add important information to understand sexual transmission of HIV from men to women. However, a number of major and minor issues need to be addressed regarding data presentation and analysis. Many of the comparisons are not shown and there are multiple inconsistencies between the figures, legends and interpretation. Please see specific comments below.

- Figure 2: Fig 2A-2D were compared with Wilcoxon Signed Ranked test, which is used to compare two groups. However, the figure shows comparison of three groups. How was this corrected for multiple comparisons? This same comment applies to figures 4 and 6. For Fig. 2E-2F the X axis needs to be defined. It would also help if 1h and 72h labels were included in the figure itself. The figure legend mentions a 7h time-point, but that is not shown (was not included in the study according to the description in results), please remove. The results section for this figure (page 7, line 143) indicate that no changes were found in IL-8, but Fig 2H shows a significant upregulation of IL-8.
- The results portion addressing BV refers to data that is not shown. This data needs to be shown, at least in supplementary format. Page 8, line 159, says women with BV (n=5) had significantly higher baseline levels of ...Compared to what? I assume women without BV in the condomless sex group, but that should be stated (and shown). The results with IL1a and sE-cad are very interesting but not discussed. A paragraph should be added to the discussion about these findings.
- The section about semen parameters (page 8, from line 178) again mentions results that are not shown. They need to be shown. Why was sE-cad not included in the repeated measures ANOVA analysis? Why was it analyzed separately and with a completely different statistical method?
- Male partner circumcision status section (page 9, from line 200). Differences in cytokines based on partner circumcision status prior to sex are not shown (line 204) and cannot be inferred from the normalized figure 3. Multiple cytokines are mentioned as not changed (line 211) but in the figure there are clear changes according to status (MIG, IFN α 2, MMP9 and potentially MIP1b). Are there any differences in sE-cad levels in semen from circumcised and uncircumcised men that could contribute to the differences observed in female genital samples?
- Fig 3. The figure legend should include how Δ was calculated. In figure legend, H and I are missing the " Δ " before the cytokine. According to figure legend, statistical analysis was Wilcoxon Signed Ranked test, is that correct? From the figure it seems like a one sample test was applied. What groups were compared? Was this comparison a one sample test compared to a hypothetical value? This information should be clarified.
- Page 10, line 216. Ref 12 is for rectal transmission, not vaginal transmission. Maybe human studies, including work from this same group and others should be mentioned here for preferential HIV target T cells.
- Fig 4. How was the CCR6 gate determined? Were FMOs or isotype controls used? These controls are needed to validate the gating strategy and the proportion of Th17 cells found. Same comments apply to Fig. 5 for activated T cells (HLA-DR) and Tcm (CCR7), which are not shown.
- Fig.5: The use of median of % of change is confusing. Is it the best way to quantify the results? How this percentage was obtained needs to be clarified in fig legend.
- Fig 5C and 5D: why are less participants included in this figure?
- Page 12, line 257: data not shown again. If this difference is from figure 6A, comparing baseline in each group, then additional multiple comparisons should be taken into consideration for the analysis.
- Fig 6 legend: A-G are not representing " Δ " as indicated in the figure legend.
- The methods section indicates that CCR5 was included in the flow panel. Were any differences detected for CCR5+ T cells? This would add very valuable information to the manuscript.

- Discussion, page 13 line 279, again ref 12 is for rectal transmission. Additional references should be included for genital transmission.
- Introduction: Line 57, references 10-13 all refer to animal studies. The authors should include some in vitro human studies defining preferential target cells for HIV.
- Line 60: MIP-3a is CCL20, not CCL22. Please correct.

Reviewers' comments:

Reviewer #1 (Remarks to the Author):

This is excellent study that adds critical information to our understanding of the dynamics of HIV transmission in cervicovaginal tissue. The authors first determine the cytokine profile of semen and cervicovaginal secretions and show that IL-1a, IL-17, IL-6, IL-8, MMP9 are enriched in cervicovaginal and IP-10, MIG, sE-cad and IFN α 2a enriched in semen. They then show that vaginal levels of IP-10, IL-1a, MIG and MIP-1b increased immediately after sex, regardless of condom use. This is an important observation as IP-10 and IL-1a cytokines have been shown to predict HIV acquisition in women. Interestingly, other proinflammatory cytokines (IL-6) were not elevated post intercourse however evidence of epithelial barrier disruption was shown. As you might expect women with bacterial vaginosis had higher levels of baseline proinflammatory cytokines and lower IL-10. Interesting however these cytokines decreased in BV+ women post condomless sex and lower cervicovaginal IL-10 levels could be correlated to levels in semen and correlated with lack of circumcision. They next show that Th17 cells and dendritic cells are increased in cervix post intercourse which are known key HIV target cells as well as activated and central memory CD4 T cells. Finally, they show that changes are strongly correlated with a lack circumcision which is a very important finding.

Thank you for finding our study important for the field.

1- Line 56-57. Langerhans cells should be added to the list of HIV target cells. These are no longer considered a dendritic cell subset and should be defined as their class of mononuclear phagocyte.

The Langerhans cells and relevant references are now added as requested (Lines 56-57).

2- Also, there are several recent studies that show that DCs can transmit HIV CD4 T cells within 1-2 hours (e.g. PMID: 33846309, 31227717, 21738469) of virus capture and Th17 cells are

present in genital tissues. Therefore, dendritic cells (DCs) probably transmit the virus to CD4 T cells within genital tissues as well as secondary lymphoid organs.

Thank you for the excellent point. This information and the related references have now been added to the Introduction (Line 57).

3- I think the baseline cytokine levels of cervicovaginal secretions vs semen are of key interest. The authors might like to consider moving this from supplemental data into the main body of data.

Thank you for your suggestion. This figure has now been moved to the main manuscript (Figure 2).

4- The flow cytometry gating strategy that defines Th17 T cells as CD3+CD4+CCR6+ cells is unconvincing. CCR6 is a tissue homing marker and this population is likely to also contain many other T cell subsets. ROR γ T should have been included to define Th17 cells. Although most cells are probably Th17 cells CXCR3 needs to be included to exclude Th1 and CCR10 to exclude Th22 cells. Furthermore, gating strategy for effector, TEM and TCM is not shown. It is important the authors present this.

Thank you. While it is true that not all CCR6+ T cells are Th17 cells, work from Tom Hope's group has demonstrated that CCR6+ T cells constitute the majority of cells infected with SIV within 12 hours after virus challenge (Stieh. 2016, Cell Host Microbe). This was the specific rationale for us to examine the impact of sex on CCR6+ T cell frequencies; the number of flow channels and cells available to us meant that we were not able to subdivide these cells further. We have clarified in the study limitations that not all these cells are Th17 cells, and that a subset represents Th22 or Th1/Th17 cells (Lines 361-365).

The gating strategies for TEM and TCM have now been added as supplementary figures (Supplementary Figure 3).

5- The flow cytometry gating strategy (supp Figure 2 not 1 as stated) for cervical DCs however is excellent and relevant to the most recent literature that shows the CD14⁺ CD11c⁺ cells are DCs and CD14⁺CD11c⁻ cells are macrophages (PMID 33846309). Nevertheless, the authors should refer to these as monocyte derived DCs as these are derived from blood monocytes as opposed to bona fide CD14⁻ CD11c⁺ DCs derived from bone marrow. They should refer to the CD14⁺CD11c⁺ DCs as cDC2.

We now refer to CD14⁺DCs as monocyte-derived DCs throughout the manuscript.

Reviewer #2 (Remarks to the Author):

In this study the authors investigated the effects of penile-vaginal sex on genital inflammation and potential epithelial barrier alterations by measuring cytokine/chemokines, immune cell populations and soluble E-cadherin in genital samples before and after sex. The study addresses an important gap in knowledge and the findings will add important information to understand sexual transmission of HIV from men to women. However, a number of major and minor issues need to be addressed regarding data presentation and analysis. Many of the comparisons are not shown and there are multiple inconsistencies between the figures, legends and interpretation. Please see specific comments below.

Thank you for the kind comment.

1- Figure 2: Fig 2A-2D were compared with Wilcoxon Signed Ranked test, which is used to compare two groups. However, the figure shows comparison of three groups. How was this corrected for multiple comparisons? This same comment applies to figures 4 and 6.

A) The p-value shown represents the results of a Wilcoxon Signed Ranked test compared to baseline for each participant.

B) We predefined IP-10 and IL-1 α levels as our co-primary endpoints for the SECS study, based on human studies of HIV acquisition and microbiome impact (referenced in the manuscript); however, since this was a pilot study, we also explored other secondary

endpoints. These analyses should be considered exploratory and confirmed in future studies. For this reason, we opted not to correct for multiple comparisons, but instead to present significant findings while emphasizing their exploratory nature in the Methods section (Lines 475-479).

- For Fig. 2E-2F the X axis needs to be defined. It would also help if 1h and 72h labels were included in the figure itself. The figure legend mentions a 7h time-point, but that is not shown (was not included in the study according to the description in results), please remove.

The 7 hour time point is now removed from the legend, and 1hr and 72 hrs labels have been added to the figure. The X axis is defined.

- The results section for this figure (page 7, line 143) indicate that no changes were found in IL-8, but Fig 2H shows a significant upregulation of IL-8.

The reviewer is correct that a change in IL-8 was seen, albeit only in the condom-protected group. This information has now been added to the Results section (Lines 146-147).

2- The results portion addressing BV refers to data that is not shown. This data needs to be shown, at least in supplementary format. Page 8, line 159, says women with BV (n=5) had significantly higher baseline levels of ... Compared to what? I assume women without BV in the condomless sex group, but that should be stated (and shown). The results with IL1a and sE-cad are very interesting but not discussed. A paragraph should be added to the discussion about these findings.

We are certainly glad to see that the reviewer finds these data interesting, although we feel that our sample size of women with BV is too small to draw firm conclusions. Therefore, we have added the data as a supplementary figure (Supplementary Figure 2) and added a sentence in the discussion highlighting that this represents an interesting area for future research (Lines 321-325).

3- The section about semen parameters (page 8, from line 178) again mentions results that are not shown. They need to be shown. Why was sE-cad not included in the repeated measures ANOVA analysis? Why was it analyzed separately and with a completely different statistical method?

The analysis referred to assessed changes in the levels of IP-10, MIG and soluble E-cadherin from baseline to one hour and 72 hours after sex. Since semen was enriched for several cytokines (MIG, IP-10 and sE-cad) relative to CVS (Figure 2), we hypothesized that semen exposure might have influenced the vaginal cytokine changes observed after condomless sex. This was not deemed to be plausible for sE-cadherin, since the post-sex increase seen in the genital tract of BV-free women far exceeded sE-cad levels present in the semen of their male partners ($median_{CVS\ 1hr} = 1,132,390.43\text{ pg/ml}$; $median_{semen} = 286,592.44\text{ pg/ml}$, respectively; $p=0.001$, Supplementary figure 1). However, since semen levels of IP-10 and MIG were much higher than those seen in CVS immediately after sex (Supplementary figure 1), it was felt that semen could be an important contributor to these changes. Therefore, we performed a repeated measures ANOVA on the change in log10 transformed vaginal concentration of IP-10 and MIG after sex in the condomless sex group by comparing models both without (ANOVA) and with baseline semen cytokine concentration as a covariate (ANCOVA), since the latter will control for exposure to IP-10 and MIG in semen. In the non-covariate model, there was a significant within person linear decrease in CVS cytokines (IP-10 and MIG) over time ($F_{(2, 50)} = 9.02$, $p < 0.001$; $F_{(1.7, 42.6)} = 22.98$, $p < 0.001$, respectively). However, after adjusting for baseline IP-10 and MIG in semen (ANCOVA), the within subject effects were not significant ($F_{(2, 48)} = 0.35$, $p = 0.703$; $F_{(1.8, 45.4)} = 1.77$, $p = 0.182$, respectively), indicating that the higher concentrations of IP-10 and MIG in semen were the likely explanation for the cervicovaginal changes seen after condomless sex.

This is now fully explained in the Results section (Lines 163-180).

4- Male partner circumcision status section (page 9, from line 200). Differences in cytokines based on partner circumcision status prior to sex are not shown (line 204) and cannot be inferred from the normalized figure 3.

The reviewer makes a good point. While prior studies have found higher cytokine levels in the coronal sulcus of the uncircumcised penis, this was not the case for our cohort (Mohammadi et al, PLOS Path 2022). Therefore, we have now removed the statement regarding differences in penile cytokine levels based on circumcision status. We have also added Supplementary Table 3, which shows pre-sex levels of CVS cytokines based on male partner circumcision status.

- Multiple cytokines are mentioned as not changed (line 211) but in the figure there are clear changes according to status (MIG, IFN α 2, MMP9 and potentially MIP1b).

The reviewer is correct that the levels of several cytokines did increase early after sex in the CVS of women with a circumcised male partner, particularly MIG. Our Results section has now been modified accordingly (Lines 208-224).

- Are there any differences in sE-cad levels in semen from circumcised and uncircumcised men that could contribute to the differences observed in female genital samples?

No difference was seen in semen sE-cad levels based on circumcision status. Similar comparisons for all immune parameters are now shown in Supplementary Table 2.

5- Fig 3. The figure legend should include how Δ was calculated. In figure legend, H and I are missing the “ Δ ” before the cytokine.

We have now added these methods to the legend and corrected H and I.

- According to figure legend, statistical analysis was Wilcoxon Signed Ranked test, is that correct? From the figure it seems like a one sample test was applied. What groups were compared? Was this comparison a one sample test compared to a hypothetical value? This information should be clarified.

The Wilcoxon Signed Ranked test was used to prepare this figure. The participants were divided into circumcised and uncircumcised groups, and a paired analysis was then used to compare the change from baseline in cytokine levels at 1hr and 72hrs respectively. We have now reworded the results to clarify that we did not directly compare the degree of change between participants who had a circumcised versus uncircumcised partner (Lines 210-224), as our study was underpowered for such an analysis.

6- Page 10, line 216. Ref 12 is for rectal transmission, not vaginal transmission. Maybe human studies, including work from this same group and others should be mentioned here for preferential HIV target T cells.

Thank you, other relevant references are now added to this line (Line 230).

7- Fig 4. How was the CCR6 gate determined? Were FMOs or isotype controls used? These controls are needed to validate the gating strategy and the proportion of Th17 cells found. Same comments apply to Fig. 5 for activated T cells (HLA-DR) and Tcm (CCR7), which are not shown.

An FMO control was used to set the gating for the CCR6+ T cells and has now been added to Figure 5 and clarified in the Methods section (Lines 445-446). The gating strategy for activated T cells and Tcm has now been added to the supplementary materials (Supplementary Figure 3).

8- Fig.5: The use of median of % of change is confusing. Is it the best way to quantify the results? How this percentage was obtained needs to be clarified in fig legend.

The change in the percentage of each cell subset was calculated from baseline, and the median change is presented in this figure. The description is now added to the legend. Combining these data in one figure was felt to be the best way to summarize these results.

9- Fig 5C and 5D: why are less participants included in this figure?

Three participants in the condomless group had positive PSA results at the 72 hour visit and were excluded from analysis at this timepoint since it is likely that they had had unprotected sex during the interval. In addition, one participant in the condom-protected sex group missed their 72 hour visit. The rationale is described in the Methods, and this information is now presented in the Results section (Line 100-105).

.
- Page 12, line 257: data not shown again. If this difference is from figure 6A, comparing baseline in each group, then additional multiple comparisons should be taken into consideration for the analysis.

Rather than adding additional data to the paper, for the sake of simplicity we have opted to remove this sentence.

10- Fig 6 legend: A-G are not representing “ Δ ” as indicated in the figure legend.

- The methods section indicates that CCR5 was included in the flow panel. Were any differences detected for CCR5+ T cells? This would add very valuable information to the manuscript.

The error in the legend has now been corrected. Thank you for your suggestion regarding CCR5. There were no significant changes in the number and percentage of CD4+CCR5+ T cells after condomless sex; however, the percentage of these cells increased 72 hours after condom-protected sex. This information has now been added to the Results section (Lines 247-250) and is included in Figure 6.

11- Discussion, page 13 line 279, again ref 12 is for rectal transmission. Additional references should be included for genital transmission.

Thank you, relevant references have now been added.

12- Introduction: Line 57, references 10-13 all refer to animal studies. The authors should include some in vitro human studies defining preferential target cells for HIV.

Thank you, the relevant references have now been added (Line 57).

13- Line 60: MIP-3a is CCL20, not CCL22. Please correct.

This has now been corrected (Line 61).

REVIEWERS' COMMENTS:

Reviewer #1 (Remarks to the Author):

All my concerns have been adressed. Congratulations on this excellent study.

Reviewer #2 (Remarks to the Author):

The authors have addressed all the concerns.